# Characterization of the Cellular Reaction to a Collagen-Based Matrix: An In Vivo Histological and Histomorphometrical Analysis

**DOI:** 10.3390/ma13122730

**Published:** 2020-06-16

**Authors:** Samuel Ebele Udeabor, Carlos Herrera-Vizcaíno, Robert Sader, C. James Kirkpatrick, Sarah Al-Maawi, Shahram Ghanaati

**Affiliations:** 1Department for Oral, Cranio-Maxillofacial, and Facial Plastic Surgery, Frankfurt Orofacial Regenerative Medicine (FORM) Lab, Johann Wolfgang Goethe University, 60590 Frankfurt am Main, Germany; seudeabor@kku.edu.sa (S.E.U.); maxilofacialchv@gmail.com (C.H.-V.); robert.sader@kgu.de (R.S.); kirkpatrick@uni-mainz.de (C.J.K.); sarah.al-maawi@kgu.de (S.A.-M.); 2Department of Oral and Maxillofacial Surgery, College of Dentistry, King Khalid University, Abha 61471, Saudi Arabia

**Keywords:** cellular reaction, collagen-based matrix, Mucomaix^®^ matrix, histology, histomorphometry, liquid PRF

## Abstract

The permeability and inflammatory tissue reaction to Mucomaix^®^ matrix (MM), a non- cross-linked collagen-based matrix was evaluated in both ex vivo and in vivo settings. Liquid platelet rich fibrin (PRF), a blood concentrate system, was used to assess its capacity to absorb human proteins and interact with blood cells ex vivo. In the in vivo aspect, 12 Wister rats had MM implanted subcutaneously, whereas another 12 rats (control) were sham-operated without biomaterial implantation. On days 3, 15 and 30, explantation was completed (four rats per time-point) to evaluate the tissue reactions to the matrix. Data collected were statistically analyzed using analysis of variance (ANOVA) and Tukey multiple comparisons tests (GraphPad Prism 8). The matrix absorbed the liquid PRF in the ex vivo study. Day 3 post-implantation revealed mild tissue inflammatory reaction with presence of mononuclear cells in the implantation site and on the biomaterial surface (mostly CD68-positive macrophages). The control group at this stage had more mononuclear cells than the test group. From day 15, multinucleated giant cells (MNGCs) were seen in the implantation site and the outer third of the matrix with marked increase on day 30 and spread to the matrix core. The presence of these CD68-positive MNGCs was associated with significant matrix vascularization. The matrix degraded significantly over the study period, but its core was still visible as of day 30 post-implantation. The high permeability and fast degradation properties of MM were highlighted.

## 1. Introduction

Many formulations of collagen-based matrices are currently in use for various soft tissue augmentations and regenerative procedures in medicine and dentistry [1,2,3,4]. Each of these products according to the manufacturers holds the promise of being suitable and functional alternative for autogenous soft tissue grafts, which to date remains the “gold standard” that these other products must meet or even surpass in properties [1,2,3,4]. Some of these desirable properties include the ability to induce tissue regeneration, act as scaffold for conduction of newly formed tissue, encourage adequate soft tissue remodeling, present good esthetics and avoid immunologic reactions with the host tissue [1]. However, most of the available matrices do not meet all of these expectations, therefore necessitating a continued search for an ideal alternative.

Although autografts are considered to be ideal for several intraoral soft tissue augmentations, they are fraught with various setbacks to their uses. These range from limited availability, morbidity of donor site, the risk of necrosis of the transplanted mucosa, color difference, and in some cases the need for secondary surgery [4,5,6]. An attempt also at culturing oral epithelial cells for intraoral soft tissue and mucosal grafting was previously made without much success [7], as this has remained only in the in vitro stage without clinical application.

Many alternatives to autografts have therefore been developed and are currently in use because of these setbacks. Different materials originating from allografts, xenografts and alloplasts have all been used for intraoral soft tissue replacements with varying degrees of success [3,6,8,9,10].

Collagen membranes and matrices are xenografts sourced mainly from tendon, dermis, skin and pericardium of bovine or porcine origin [11]. Collagen material for use as membranes and matrices has several advantages. These include ready availability, ease of manipulation, low antigenicity, hemostatic and wound healing properties [1,2,12]. They can be developed from type I collagen or a combination of types I and III collagen [2]. They are readily biodegradable which means there is no need for a second surgery for their removal especially when used as membranes in guided tissue regeneration (GTR) [1,2,3]. However, the ease of biodegradation of collagen means that they lack the membrane stability required to function as good barriers in GTR [3].

In order to increase the mechanical stability of collagens and prolong their resorption, chemical cross-linking was introduced. This was achieved by various techniques such as the use of ultraviolet light, glutaraldehyde, diphenylphosphorylazide, or hexamethylenediisocyanate [13]. However, even though cross-linked collagens are more stable mechanically, they were shown to prevent the attachment and proliferation of periodontal ligament fibers and osteoblasts in humans when compared to non-cross-linked collagens [13]. This therefore led to the development of other collagen membranes with a different chemical processing. In some instances, non-cross-linked, easily degradable type III collagen was combined with more stable type I collagen [3].

The combination process according to some authors helps to regulate their bioresorbability; allowing the material to persist for a longer period in the implantation bed and eliciting mild inflammation [3,14]. However plausible this may sound, it is noteworthy that different formulations of this non-cross-linked collagen currently in use have so far yielded conflicting results in terms of rate of biodegradation [15].

In comparison with the cross-linked membranes, a recent systematic review comparing cross-linked to non-cross-linked collagens in terms of their effects on bone, yielded no significant difference except for the higher rates of post-operative complications associated with the cross-linked variants [16]. 

The type of cellular reactions induced by different formulations of collagen matrix determines their suitability or otherwise as barriers and placeholders in various surgical augmentations and GTR. Non-cross-linked collagen matrices for example, have been shown to induce mainly mononuclear cells when implanted in in vivo settings, thus eliciting only mild inflammatory tissue reaction without the involvement of multinucleated giant cells (MNGCs) [3,15]. Although the presence of MNGCs would enhance implantation site vascularization, they may contribute to a faster degradation and sometimes disintegration of the implanted biomaterial [17,18].

The importance of intraoral soft tissue augmentation cannot be overemphasized as it cuts across different aspects of clinical dentistry. This ranges from providing for soft tissue coverage following various surgical resections [4]; improving aesthetics in areas of gingival recessions [6]; allowing for a stable soft tissue base for the insertion of different shapes and designs of dental implants [19] and orthodontic miniscrews [20] among others.

This study therefore evaluated the inflammatory tissue reaction to Mucomaix^®^ matrix (MM), a non-cross-linked collagen-based matrix with an open, interconnecting porous structure in an in vivo setting using Wistar rats. The rate of degradation, level of integration to the implantation site, the induced cell types and percentage vascularization of the embedded matrix were assessed in vivo. The permeability of this matrix and its initial reaction to mononuclear human cells were also assessed ex vivo using liquid platelet-rich fibrin (liquid PRF). These were intended to show its efficacy and adaptability in human tissue.

## 2. Materials and Method

### 2.1. Mucomaix^®^ Matrix

Mucomaix^®^ matrix MM, (Matricel Gmbh, Herzogenrath, Germany) is a resorbable matrix composed of highly purified porcine collagen and elastin fibers. The materials for its production are harvested in the EU under strict veterinary control. It is CE-marked and FDA approved. It is made from collagen types I and III without chemical cross-linking. MM is double-packaged and sterilized by ethylene oxide gas treatment. It is effectively purified and is therefore highly biocompatible and considered safe with regard to potential viral transmission [21].

MM has an open, interconnecting porous structure (Figure 1), which according to the manufacturers, provides a matrix for the migration of proliferating cells and vascular structures. They also claimed that as the healing process advances, MM is degraded whilst a new soft tissue matrix is regenerated inside the matrix structure. MM is hydrophilic and retains its structural integrity also in the wet state [21].

### 2.2. Ex Vivo Study with Liquid Platelet-Rich Fibrin (Liquid PRF)

This ex vivo aspect of the study is crucial in order to assess the biomaterial membrane capacity to absorb human proteins and their interactions with mononuclear cells from peripheral blood. A reliable method to assess this initial biomaterial–cell interaction is by the use of liquid PRF, which is a centrifuged autologous blood concentrate rich in platelets, leukocytes, and plasma proteins all suspended in a soluble fibrinogen matrix [22]. Ethical approval to carry out studies on human tissues was sought and obtained from the Ethics Committee of the Goethe University, Frankfurt (IRB No. 265/17) and the study was conducted in accordance with the Declaration of Helsinki.

Three healthy volunteers aged between 18 and 60 years who gave written informed consent, donated blood for this ex vivo study. The liquid PRF preparation was performed as previously described by our group [23,24]. Peripheral blood was collected using 10-ml plastic tubes (Orange tubes, PROCESS for PRF, Nice, France) and clinically approved butterfly needles. They were immediately placed in a pre-programed centrifuge (DUO Centrifuge, PROCESS for PRF, Nice, France) and centrifugation was completed following liquid PRF protocol (10 ml, 700 rpm, 60 g, for 3 min).

The liquid PRF, which is the yellowish upper layer, was then collected into a syringe (BD Microlance^TM^ 3, Heidelberg, Germany). Three biomaterial samples per donor measuring 10 × 10 mm^2^ were placed in a cell culture plate and 1ml of liquid PRF added to each and incubated for 15 minutes at room temperature. The samples were then fixed for 24 h in 4% buffered formalin for further histological analysis using hematoxylin and eosin (HE) stain. This same histological analysis was also performed for untreated MM without liquid PRF to show its native structure.

### 2.3. Experimental Design of the In Vivo Study

Approval for this study was obtained from the Committee on the Use of Live Animals in Teaching and Research of the State of Hessen, Germany (No. FK/1023) and ARRIVE (Animal Research: Reporting of In Vivo Experiments) guideline for in vivo experiments was used. Twenty-four Wister rats were acquired from Charles River Laboratories (Sulzfeld, Germany) and kept in the Animal Laboratory Unit, Institute of Pathology, Goethe University Hospital Frankfurt, Germany. They were allowed an acclimatization period of one week in order to get used to the laboratory environment before the start of the study proper. The animals were fed on a regular basis with mouse pellets (Laboratory Rodent Chow, Altromin, Lage, Germany) and water was available also for them at all times (ad libitum). Artificial light–dark cycles of 12 h each simulated day and night rhythms.

The 24 animals were first divided into 2 groups of 12 animals each. One group served as the experimental group whereas the other served as the control. They were then further divided randomly into 3 groups of 4 animals each, i.e., 4 animals per time points of 3, 15 and 30 days. The experimental group had MM implanted into their subcutaneous pockets, while the control group were sham-operated without biomaterial implantation to evaluate the inflammatory pattern during wound healing. This implantation was conducted under sterile conditions following previously established subcutaneous implantation methods [3]. All animals survived the procedure until the evaluation time points without any complications. 

### 2.4. Tissue Preparation for Histology and Histochemistry

The materials were explanted at the different specific time points and the tissue samples were then prepared for histology and histochemistry following previously established protocols [3,15]. They were fixed in 4% buffered formalin for 24 h before being cut into three identical segments, which included the margins and the center of the scaffold. These samples were placed into embedding cassettes (Histosette, VWR, Darmstadt, Germany) and dehydrated in baths of progressively concentrated ethanol (70–100%) before alcohol clearance with xylene. The tissue samples were then embedded into paraffin blocks and after sufficient cooling, the paraffin blocks were cut with a rotatory microtome (Rotationsmikrotom RM2255, Leica, Nussloch, Germany) to produce serial sections of 3 μm thickness. The first tissue section was stained with Mayer’s hematoxylin and eosin (HE) stain, the second was stained using Azan trichrome stain, whereas the third and fourth sections were stained with Masson–Goldner and tartrate resistance acid phospastase (TRAP) stains, as previously described [25,26,27].

The remaining 2 sections were finally stained immunohistochemically for CD68 and α-SMA (alpha-smooth muscle actin). This was completed using the Lab VisionTM Autostainer 360-2D (ThermoFisher Scientific, Dreieich, Germany). After deparaffinization, the slides were pre-treated with citrate buffer. Proteinase K was blocked using 4% H_2_O_2_ in methanol and endogenous avidin- and biotin-binding proteins were blocked by the avidin and biotin blocking solutions (Avidin/Biotin Blocking Kit, Vector Laboratories, Burlingame, CA, USA). The first antibody used was anti-CD-68 (MCA341GA; 1:400; 30 min), anti-human α-SMA antibody clone 1A4, whereas the second antibody was goat anti-rabbit IgG-B (sc-2040, 1:200, Santa Cruz Biotechnology, Dallas, TX, USA). Subsequently, the avidin- biotin-peroxidase complex (ABC) (30 min) and the Histostain-Plus IHC Kit including AEC (20 min) were applied (ThermoFisher Scientific, Dreieich, Germany). Counterstaining was performed using Mayer’s hematoxylin. The negative control for the immunohistochemical (IHC) staining used was the absence of incubation for primary antibody, while the positive control was applied according to the manufacturer’s instruction.

### 2.5. Qualitative Histological Analysis

A Nikon ECLIPSE 80i microscope (Nikon, Tokyo, Japan) equipped with a motorized stage (ProScan III, Prior, Rockland, MA, USA) and NIS Elements software (Nikon, Tokyo, Japan) was used for qualitative histological analysis at ×200 magnification as described in previous publication [17]. This focused on the cellular reaction and inflammatory pattern towards the implanted biomaterial, vascularization of the implantation site, signs of fibrosis, encapsulation, and matrix degradation.

### 2.6. Quantitative Histomorphometric Analysis

Histomorphometry of the stained slides was performed using a Nikon ECLIPSE 80i light microscope as previously described [17]. Total scans of the sample, which included the collagen matrix and the peri-implant tissue, were reconstructed automatically by merging 100–130 individual micrographs. The thickness of the MM harvested from each animal at each of the three time points (3, 15, and 30 days) was then measured in 15 distinct points along its thickness. The mean of these measurements was calculated as the membrane thickness in micrometers. The values obtained from the later time points (i.e., 15 and 30 days) were then compared to that of day 3, which was assigned a value of 100%.

The number of the multinucleated giant cells (MNGCs) and their subpopulations (CD-68-positive and CD-68-negative giant cells) was manually counted using the annotations and measurement function of the NIS Elements software. The total number of each of these cells was then calculated in relation to the implantation area (MNGCs/mm^2^) and also as a percentage of the total cells present in a given area. These values were then compared statistically for all the 3 time points.

In the same vein, the number of vessels and the area they occupied (mm^2^) within the digitized scans were manually marked with the NIS Element software also. This then enabled the computations of the vessel density of the total scan (vessels/mm^2^) and the percentage of the vascularized area in the implantation site.

### 2.7. Statistical Analysis

The primary endpoints were the characterization of the inflammatory cellular reaction in terms of the induction of mononuclear cells, MNGCs and the vascularization pattern. The secondary outcomes were biomaterial degradation and the signaling of molecule expressions of CD68. For the qualitative histological evaluation of the ex vivo study, the number of samples (n = 3) necessary to obtain reproducibility was selected according to previous studies [28,29]. The sample size calculation for the in vivo study (n = 4) was calculated also according to previous studies, showing differences between groups when the number of animals was 4 [24,29]. The data collected from the histomorphometric studies were tested for normal Gaussian distribution using the Shapiro–Wilk normality test and statistically analyzed using the graphing and statistics software GraphPad Prism version 8 (GraphPad Software, Inc., La Jolla, CA, USA). The results were expressed as mean ± standard deviation (SD). Statistical significance was calculated using one-way and two-way analysis of variance (ANOVA) with a Tukey multiple comparisons test (α = 0.05, 95% CI for the mean difference) of all pairs. Differences were considered statistically significant when *p*-value is <0.05 and this was categorized as * = *p* < 0.05, ** = *p* < 0.01, *** = *p* < 0.001, and **** = *p* < 0.0001 based on level of significance.

## 3. Results

### 3.1. Ex Vivo Histological Analysis

The initial structure of the analyzed MM (Figure 1A,B) revealed a loosely arranged interconnected collagen fibrils with large irregular pores. The result of the ex vivo study showed that the matrix absorbed the liquid PRF with fibrin clots seen within the pores (Figure 1C,D). Additionally, the matrix was also permeable to inflammatory cells and platelets from peripheral blood, as these were seen throughout its porous structure (Figure 1C,D).

### 3.2. In Vivo Histological and Histomorphometric Analysis

#### Qualitative Analysis of Cellular Reaction to MM over the Study Period

The 24 animals used as both test and control groups survived the operations and did not show any complication throughout the entire implantation time. The MM was clearly visible in the subcutaneous implantation site of the animal test group throughout the three different time points of this study. At day 3 after implantation, the biomaterial maintained its structural integrity and elicited only a mild physiological reaction. Mononuclear cells lined the surfaces, and some of these cells penetrated the surface structure of the biomaterial (Figure 2A,A’). The control group at this stage had more mononuclear cells than the test group. There was no vascularization of the matrix structure at this time point, however, few blood vessels were seen in the implantation site (Figure 3 and Figure 4).

After 15 days of implantation, the biomaterial had significantly reduced in size in comparison to day 3 (*p* < 0.0001). However, there was only a slight increase in mononuclear cells, which accumulated on the surfaces of the biomaterial, and some were found to have invaded the biomaterial structure. They were mainly in the outer two-thirds of the MM without getting to the central region (Figure 2B,B’). It should be noted that few multinucleated giant cells (MNGC) were seen in this point of the study mainly in the implantation site and the outer part of the matrix structure. Micro-vessels were also seen in the implantation site and in the upper layer of the biomaterial.

Day 30 post-implantation showed that the matrix core was still largely intact, but there was, however, a significant reduction in the size of the biomaterial (*p* < 0.0001) (Figure 2C,C’). There was also a marked increase in mononuclear cells (CD-68-positive macrophages) both in the implantation site and within the central region of the matrix (Figure 5). The matrix at this time point is entirely embedded in a rich connective tissue with neo-vascularization. In a variation to the pattern seen at day 15, the micro-vessels were seen both at the periphery and within the core of the matrix structure (Figure 3). There was also a marked increase in MNGCs at this time point compared to day 15, both in the implantation site and within the matrix. Most of these MNGCs were CD-68-positive (Figure 5).

### 3.3. Quantitative Histomorphometric Analysis

#### 3.3.1. Evaluation of MM Degradation 

Histomorphometric analysis of the MM showed that there was a steady decrease in thickness from day 3 till day 30 post implantation (Figure 2). At day 3 post implantation, the mean thickness of the implanted biomaterial was 2663.59 μm ± 499.28, whereas days 15 and 30 were 1982 μm ± 653.3 and 767.32 μm ± 276.63, respectively. The biomaterial degradation in these two time points were highly statistically significant (*p* < 0.0001) when compared to day 3.

#### 3.3.2. Evaluation of the Total Amount of CD-68-Positive Mononuclear Cells

The density of CD-68-positive mononuclear cells was evaluated histomorphometrically. This was done by calculating the number of the positive cells per square millimeter for the three different time points of the study. The macrophage density at day 30 was significantly higher than days 3 (*p* < 0.01) and 15 (*p* < 0.01). However, there was no statistical significant increase in the density of the macrophages between days 3 and 15. The control group showed significant increase in density of macrophages when compared to the test group on day 3 (*p* < 0.01), but a reduction on day 30 (*p* < 0.05) (Figure 5 and Figure 6).

#### 3.3.3. Evaluation of Total Amount of Multinucleated Giant Cells (MNGCs)

The density also of multinucleated giant cells was evaluated histomorphometrically. None of these cells were seen as of day 3 post implantation; they were only seen as from the day 15. Day 30 post implantation showed that there was a statistically significant increase in the number of MNGCs from day 15 (*p* < 0.05). A significant number of these MNGCs seen on both days 15 and 30 were also CD-68-positive (Figure 5). These CD-68-positive MNGCs were also significantly higher on day 30 when compared to day 15 (*p* < 0.05).

#### 3.3.4. Evaluation of Implantation Site Vascularization 

Histomorphometric analysis of the vascularization pattern showed that there was a steady increase in percentage of vascularization of the implantation site from day 3 onwards. Day 3 post implantation revealed vascularization only in the implantation site, which are believed to be native blood vessels. There was an increase in vascularization percentage on day 15 in comparison to day 3. These vessels were located mainly in the implantation site and the periphery of the biomaterial structure. Additionally, day 30 percentage of vascularization was significantly higher than day 3 (*p* < 0.05). It is noteworthy that only on day 30 post implantation, was the percentage of vascularization of the MM group significantly higher than that of the sham-operated group (*p* < 0.05).

There was also a marked increase in vessel density (number of vessels detected per square millimeter) of the implantation site at day 30 compared to day 3 (*p* < 0.05) (Figure 3 and Figure 4).

## 4. Discussion

This study primarily focused on ex vivo and in vivo evaluation of the tissue response to MM, a non-cross-linked collagen-based matrix, in terms of its permeability, level of integration to the implantation site, rate of degradation, the induced cell types and percentage of vascularization of the embedded matrix. Comparisons were also made to the control group, which are sham-operated without biomaterial implantation just to simulate normal physiologic wound healing process.

The ex vivo aspect of this study used liquid PRF to simulate the initial interactions of the biomaterials with the blood cells. The liquid PRF is a centrifuged autologous blood concentrate rich in platelets, leukocytes, and plasma proteins, all suspended in a soluble fibrinogen matrix [22]. In the present study, the biomaterial absorbed the liquid PRF and the cells (leukocytes and platelets) could be seen everywhere within the entire substance of the matrix. This goes to show the highly porous nature of this collagen matrix and its permeability to human cells. Similar ex vivo analysis had previously been done for some biomaterials by our group with varying results [30,31]; while some differentially absorbed the PRF, others were impermeable. A collagen membrane of bovine origin [28] showed a similar pattern to MM in terms of its interaction with PRF; it allowed initial infiltration of its structure by the liquid PRF. This, however, contrasted the initial results obtained when a sugar-coated cross-linked collagen membrane was analyzed ex vivo. It was practically impervious to the liquid PRF, preventing cellular migration across its membrane [24]. The degree of porosity of these biomaterials, their hydrophilic nature and surface polarity seem to be major factors in the absorption of the liquid PRF [24,28]. It therefore means that the biomaterials characteristics and surface properties are crucial for the interaction with PRF [28]. However, further studies are necessary to elucidate the main reasons. The results of the present study therefore suggest that this matrix would be able to act as a suitable scaffold and a carrier. This allows cellular migration within its structure and, subsequently, may be able to encourage tissue regeneration.

There was a significant reduction in the thickness of the biomaterial over the 3 time points of this study, which are days 3, 15 and 30. This reduction in thickness was steady with no disintegration and the matrix core was still visible as of day 30 post-implantation. This was similar to what was observed with non-cross-linked collagen types I and III as previously demonstrated by our group [3]. A fast degradation may not be beneficial for any biomaterial that is to serve as a good placeholder [32]. A longer implantation period may therefore be necessary to objectively evaluate this property for MM. Additionally, the fast degradation of the collagen matrix in this present study also attests to its very porous nature and high permeability.

On the other hand, it was observed in a previous study that ethylene oxide cross-linked collagen type I and III showed a long degradation period and higher foreign body tissue reaction. This slow integration process and the induction of high tissue inflammatory reactions associated with chemical cross-linking means they are less suitable for soft tissue GTR with the possibility of wound-breakdown and other soft tissue complications [32,33]. It therefore means that different formulations of collagen biomaterials should be tailored to different clinical indications based on their cellular reaction and degradation pattern.

The matrix under study elicited only a mild physiological reaction on day 3 post implantation. Few inflammatory cells were seen on the surfaces of the biomaterial, with a few penetrating its surface structure at this time point. The control group at this time also had significantly higher number of inflammatory cells than the test group. There was, however, a slight increase in the density of these cells on day 15 post implantation than day 3, but it was still not statistically significant. The only difference was that some of these cells had penetrated deeper into the substance of the biomaterial and occupied the outer two-third of the collagen matrix. Only on day 30 was there significant rise in the density of the inflammatory cells when compared to days 3 and 15. These cells were mostly CD-68-positive macrophages. Macrophages are known to be among the first line of immune cells that interact with implanted biomaterials and they play crucial roles in the biomaterial vascularization, integration to the implantation site and degradation patterns [34]. These roles are basically influenced by the local environment and the surface structure of the implanted biomaterial that causes the polarization of the macrophages into two different phenotypes; the classically activated M1 and the alternatively activated M2 variants [34].

Multinucleated giant cells (MNGCs) were seen in the implantation site and within the biomaterial structure as from day 15 onwards. Their number significantly increased on day 30 post-implantation and most of them were also CD-68-positive. These MNGCs are suggested to be partly formed by the fusion of initially aggregated macrophages in response to the foreign body (biomaterial) in order to enhance phagocytosis [35]. The nature and roles of these cells in biomaterial degradation and integration to the implantation site have been a controversial issue [36,37,38]. A clear understanding of their functions and proper phenotypic characterization into the pro-inflammatory M1-MNGCs or the wound-healing M2-MNGCs is vital to the development of more biocompatible biomaterials [30]. Our group has previously demonstrated that most of the MNGCs around xenogeneic bone materials were mostly pro-inflammatory as they expressed CCR7 and Cox-2 [38]. Pro-inflammatory macrophages and MNGCs are known to be actively involved in biomaterial degradation and vascularization [36]. These properties are enhanced by their ability to express very important growth factors and cytokines like the vascular endothelial growth factor (VEGF) necessary for angiogenesis [34,36]. The full characterization of biomaterial-induced MNGCs into the pro-inflammatory and anti-inflammatory phenotypes for different biomaterial subtypes and their roles in tissue regeneration deserves further evaluation and is a current research topic by our group.

The density of MNGCs and the percentage vascularization seems to be directly correlated to the degradation pattern of implanted biomaterials [38]. As previously documented in case of synthetic biomaterials, significantly higher number of MNGCs and higher rate of vascularization were induced, thereby leading to faster biomaterial degradation and in some cases disintegration [35,38]. On the other hand, xenogeneic biomaterials induced a milder MNGC accumulation and a lower percentage of vascularization, leading to a more gradual degradation and more time for newly formed tissue ingrowth and integration [35,38].

Based on the induction of MNGCs by different biomaterials, our group recently proposed the classification of these biomaterials into three distinct groups [39]. Class I represents those biomaterials that does not induce MNGCs throughout the implantation period (e.g. Biogide^®^, Mucograft^®^); class II represents those biomaterials that induce MNGCs as from day 3 onwards and then maintain a constant level from around day 15 to day 30 (e.g., Bego^®^ collagen membrane, Ossix^®^ plus); and class III are those that show increasing tendency of MNGCs induction up until day 30 post-implantation (e.g., Collprotect^®^, Mucoderm^®^). MM clearly belongs to class III as it induced an increasing number of MNGCs up until day 30 post-implantation in addition to a significant increased vascularization. This may have accounted for the marked reduction in the size of the matrix as seen in day 30. Although there was no disintegration of the matrix at the last explantation date (i.e., 30 days), the rate of degradation was so fast that the entire matrix may actually completely degrade and disappear from the implantation site if the study prolonged further.

In a split-mouth clinical study using human tissue biopsies and comparing tissue reactions to a synthetic bone substitute (Nanobone) and a xenogeneic bone substitute (Bio-oss), our group had previously demonstrated that the induced MNGCs are foreign body giant cells without osteoclastic functions. This is because even their increased presence with associated increased vascularization did not lead to more tissue or bone regeneration [40].

The present study revealed that there was an increase in vascularization percentage on day 15 in comparison to day 3, even though this was not statistically significant. These vessels were located mainly in the implantation site and the periphery of the biomaterial structure. However, day 30 percentage vascularization was significantly higher than day 3, and in this case the vessels penetrated to the core of the biomaterial. This pattern of vascularization and cell invasion was similar to what was earlier described for native, non-cross-linked collagen matrices. Large areas of the collagen being invaded by blood vessels and mononuclear cells after just 2 weeks of implantation and, then, a homogenous invasion of the entire scaffold structure by connective tissue after four weeks [32]. Different collagen biomaterials elicit different patterns of vascularization: while some elicit high vascularization and therefore significantly higher connective tissue ingrowth, others elicit only minimal vascularization [15]. It therefore raises the question as to what level of vascularization is necessary for a successful integration and new tissue regeneration. This was the point of study by Ghanaati et al [15] in which they demonstrated that some collagen membranes do not require high vascularization rates for their successful integration.

## 5. Clinical Relevance

Non-cross-linked collagen matrix has found clinical use in augmentations of both intra oral and extra oral defects, improving both functional and esthetic outcomes [3,41,42,43]. Our group had previously demonstrated the successful use of a bilayered, non-cross-linked collagen matrix to augment extra oral defects following cancer excision. All the defects were covered by newly regenerated skin, which bears similar resemblance to the surrounding skin in terms of color and texture [41]. Additionally, another multicenter case series reported by Laino et al showed acceptable outcome with the use of non-cross-linked collagen matrix to cover intra oral excisional biopsy defects [42].

Even though MM as an example of non-cross-linked collagen matrix may show a fast degradation as seen in the present study, it lasts long enough to allow newly formed connective tissue ingrowth into the matrix scaffold. This therefore implies that it may find use in soft tissue augmentations in some selected cases. However, further clinical studies would be necessary to show the suitability of this matrix as a functional alternative to autogenous soft tissue grafts. 

## 6. Conclusions

This study analyzed the ex vivo and in vivo cellular reactions to MM, a non-cross-linked collagen matrix made from porcine collagen and elastin fibers. The ex vivo aspect of the study showed the highly porous nature of the matrix as it allowed the penetration of its structure by blood cells and plasma proteins when immersed in liquid PRF. Analysis of the in vivo cellular reaction was also completed using subcutaneous implantation in Wister rats. This revealed a mild tissue inflammatory reaction as of day 3 post-implantation with the presence of mononuclear cells, which were mostly CD-68-positive macrophages. These cells were mainly aggregated in the implantation site, and some in the surface structure of the biomaterial. From day 15 onwards, MNGCs were seen in the implantation site and within the outer third of the matrix. Their number significantly increased on day 30 with a homogenous spread even to the core of the collagen matrix. Interestingly, the presence of these CD68-positive MNGCs was associated with significant vascularization and the invasion of the matrix structure with rich connective tissue. This may have accounted for the marked degradation of the matrix over the study period. The roles of these MNGCs are yet to be fully understood, and further studies would be necessary to characterize their importance in tissue regeneration. Additionally, a long term in vivo implantation period may be necessary to fully assess the degradation pattern of this collagen matrix.

## Figures and Tables

**Figure 1 materials-13-02730-f001:**
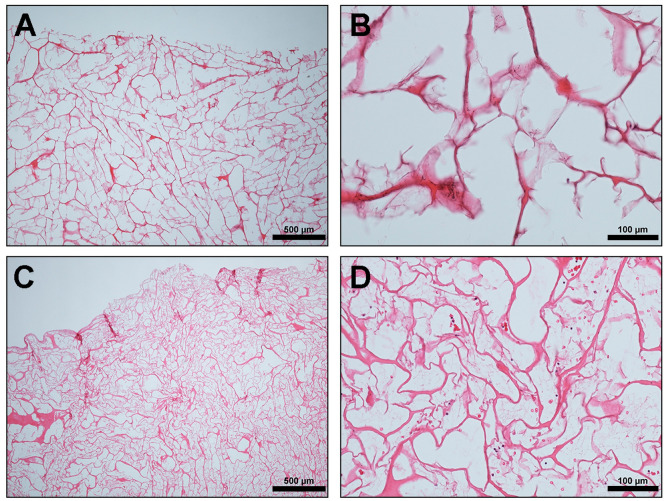
Ex vivo: (**A**,**B**) Mucomaix^®^ matrix (MM) native structure showing loosely arranged interconnected collagen fibrils with large irregular pores (hematoxylin and eosin (HE) stain; (**A**) ×4; scale bar = 500 μm and (**B**) ×20; scale bar = 100 μm). (**C**,**D**) Cross section of MM after placement in liquid platelet-rich fibrin (liquid PRF) showing inflammatory cells (black arrows) and erythrocytes (red arrows) within the entire porous structure. (HE stain; (**C**) ×4; scale bar = 500 μm and (**D**) ×20; scale bar = 100 μm).

**Figure 2 materials-13-02730-f002:**
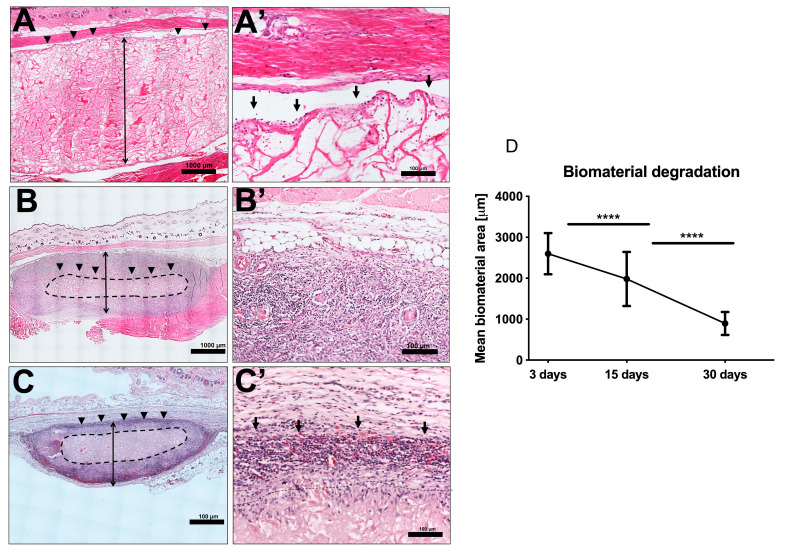
(**A**,**A’**) Day 3 post implantation showing intact matrix structure with inflammatory cells seen mainly on the interface between the biomaterial and the implantation site (black arrows). (HE stain; (**A**) Total scan; scale bar = 1000 μm and (**A’**) ×20; scale bar = 100 μm). (**B**,**B’**) Day 15 post implantation showing significant reduction in the size of the matrix with more accumulations of mononuclear cells in the implantation site and the outer third. The core of the biomaterials encircled by the black dotted lines remained free from mononuclear cells (HE stain; (**B**) Total scan; scale bar = 1000 μm and (**B’**) ×20; scale bar = 100 μm). (**C**,**C’**) Day 30 post implantation showing further degradation of the biomaterial but the core is still visible. A marked increase in mononuclear cells both in the implantation site and deeper within the matrix structure. However, the core (encircled by black dotted lines) is still free from mononuclear cells. (HE stain; (**C**) Total scan; scale bar = 1000 μm and (**C’**) ×20; scale bar = 100 μm). Blue dotted lines: total area occupied by the matrix. (**D**) Graph of histomorphometric analysis of the MM degradation pattern in vivo. Day 3 post implantation: the mean thickness of the implanted biomaterial was 2663.59 μm ± 499.28, whereas days 15 and 30 were 1982 μm ± 653.3 and 767.32 μm ± 276.63, respectively. (*p* < 0.0001) **** = *p* < 0.0001.

**Figure 3 materials-13-02730-f003:**
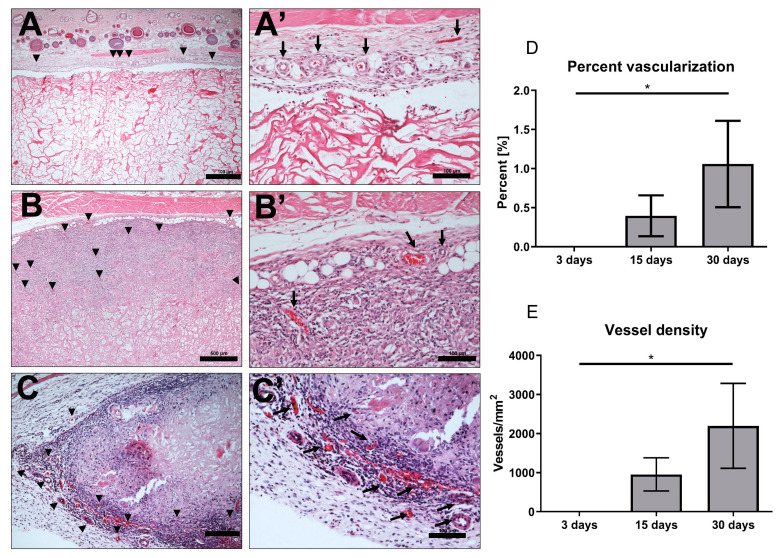
Vascularization pattern (**A**,**A’**) Day 3, showing vessels (black arrows) in the implantation site alone (H E stain; (A) ×10; scale bar = 100 μm and (A’) ×20; scale bar = 100 μm). (**B**,**B’**) Day 15, showing new vessels (black arrows) in the implantation site and the outer third of the matrix (HE stain; (**B**) ×10; scale bar = 500 μm and (**B’**) ×20; scale bar = 100 μm). (**C**,**C’**) Day 30, showing increased new vessel formation in the implantation site (black arrows) and a deeper invasion of the matrix structure (HE stain; (**C**) ×10; scale bar = 100 μm and (**C’**) ×20; scale bar = 100 μm). (**D**) Percentage of vascularization: There was a steady increase in percentage of vascularization of the implantation site from day 3 onwards. Day 30 percentage of vascularization was significantly higher than days 3 for the matrix (*p* < 0.05). * = *p* < 0.05. (**E**) Vessel Density: a marked increase in vessel density of the implantation site at day 30 compared to day 3 (*p* < 0.05).

**Figure 4 materials-13-02730-f004:**
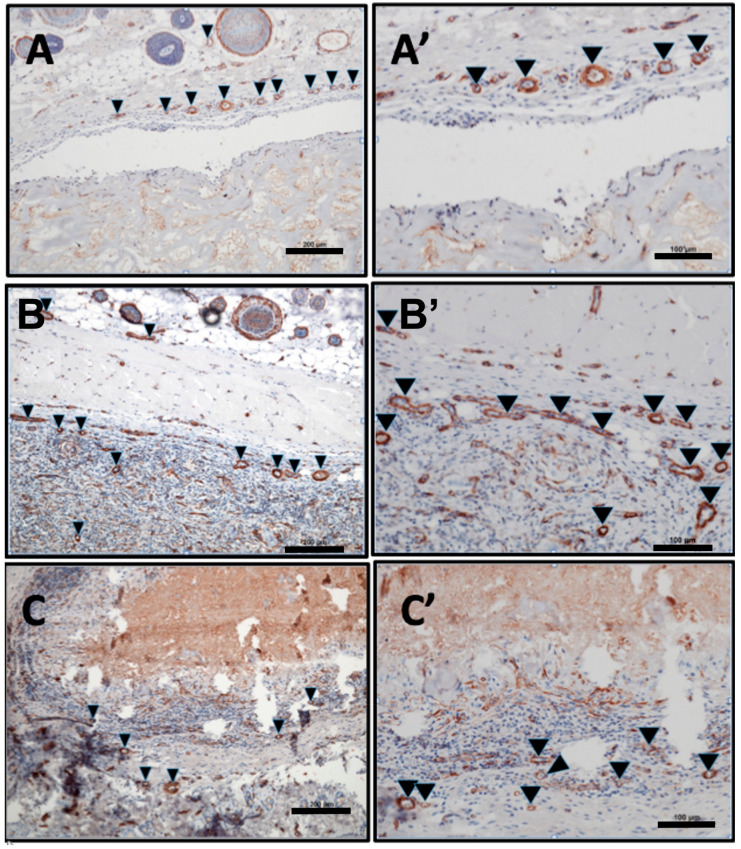
Vascularization pattern: (**A**,**A’**) Day 3, showing vessels (black arrows) in the implantation site alone (alpha-smooth muscle actin (α-SMA) stain; (**A**) ×10; scale bar = 200 μm and (**A’**) ×20; scale bar = 100 μm). (**B**,**B’**) Day 15, showing new vessels (black arrows) in the implantation site and the outer third of the matrix (α-SMA stain; (**B**) ×10; scale bar = 200 μm and (**B’**) ×20; scale bar = 100 μm). (**C**,**C’**) Day 30, showing increased new vessel formation in the implantation site and a deeper invasion of the matrix structure (black arrows). (α-SMA stain; (**C**) ×10; scale bar = 200 μm and (**C’**) ×20; scale bar = 100 μm).

**Figure 5 materials-13-02730-f005:**
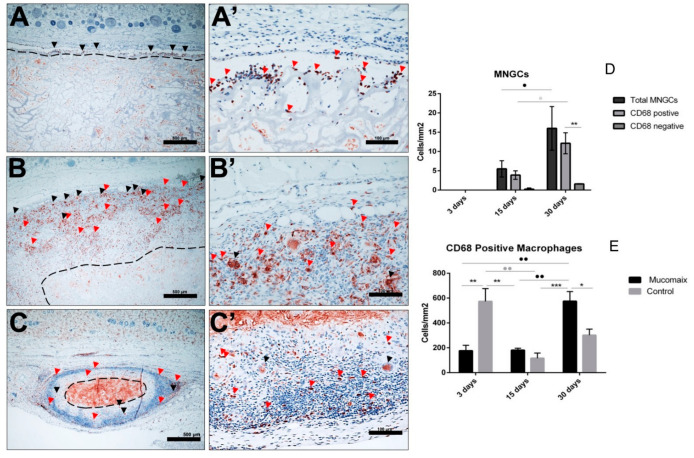
(**A**,**A**’) Day 3 with CD-68 positive macrophages lining the surfaces of the matrix and penetrating the outer part of the matrix surface structure (red arrows) (CD-68 immunohistochemical (IHC stain; (**A**) ×10; scale bar = 500 μm and (**A’**) ×20; scale bar = 100 μm). (**B**,**B’**) Day 15 post-implantation. There is an increased concentration of CD-68-positive macrophages (red arrows) and MNGCs (black arrows) both on the surface and within the outer third of the matrix structure (CD-68 IHC stain; (**B**) ×10; scale bar = 500 μm and (**B’**) ×20; scale bar = 100 μm). (**C**,**C’**) Day 30 post-implantation. Further accumulation of CD-68-positive macrophages (red arrows) and MNGCs (black arrows) with deeper penetration of the matrix structure (CD-68 IHC stain; (C) total scan; scale bar = 100 μm and (**C’**) ×20; scale bar = 100 μm). (**D**) Histomorphometric analysis of MNGCs: Significant increase in MNGCs from day 15 to 30 (*p* < 0.05). CD-68-positive MNGCs were also significantly higher on day 30 when compared to day 15 (*p* < 0.05). * = *p* < 0.05, ** = *p* < 0.01. (**E**) Histomorphometric analysis of CD-68-positive mononuclear cells: The macrophage density at day 30 was significantly higher than days 3 (*p* < 0.01) and 15 (*p* < 0.01) for the test group. The control group showed significant increase in density of macrophages when compared to the test group on day 3 (*p* < 0.01), but a reduction on day 30 (*p* < 0.05). * = *p* < 0.05, ** = *p* < 0.01, *** = *p* < 0.001.

**Figure 6 materials-13-02730-f006:**
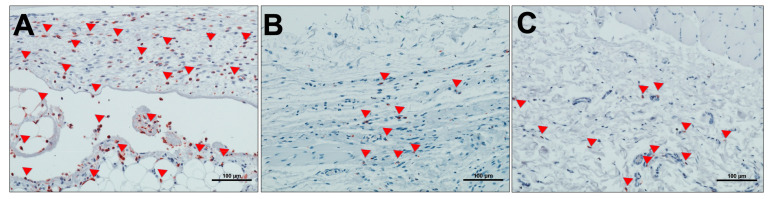
Sham-operated animal group (control) with no biomaterial implantation. (**A**) Day 3 post surgery; (**B**) day 15 post surgery; and (**C**) day 30 post surgery. Red Arrows show CD-68-positive mononuclear cells. (CD-68 IHC stain; ×200; scale bar = 100 μm). The mean area per time point calculated from the experimental groups was used to perform histomorphometry for the control group.

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
