# Peer review of "Characterization of the Cellular Reaction to a Collagen-Based Matrix: An In Vivo Histological and Histomorphometrical Analysis"

_materials, 2020, doi:10.3390/ma13122730_

Round 1

Reviewer 1 Report

The manuscript reports on the permeability and inflammatory tissue reaction to Mucomaix®, a non cross-linked collagen-based matrix. The ex vivo part showed the high porosity of the matrix as it allowed the penetration of blood cells and plasma proteins. The in vivo part showed initially a mild tissue inflammatory reaction followed by increased vascularization and presence of MNGCs.

Overall remarks
A nice study with clear results, which are well presented
Specific remarks
Liquid PRF appears in the abstract without any hint, what it is. A definition is given at the very end of the introduction.
For the discussion: is the sterilization methodology ethylenoxide responsible for the mild immunreaction?

Reviewer 2 Report

The manuscript “Characterization of the Cellular Reaction to a Collagen-based Matrix: An In Vivo Histological and Histomorphometrical Analysis” explores the influence of the commercial matrix Mucomaix® on the permeability, level of integration to the implantation bed, degradation, inflammation and vascularization. From the combination of the ex vivo and in vivo studies, clinically-relevant information is obtained, considering that this matrix is a suitable alternative for autogenous soft tissue grafts. I suggest the publishing of this manuscript as Article after Minor Revisions, which are listed below:

  1. Abstract. The acronyms PRF and MNGCs should be defined.
  2. Figure 2. The data for the mean biomaterial degradation over time differs from the figure to the legend. For example, the authors mention that at day 15 it is 2246 but in the figure seems around 2000. How can the authors explain this difference?
  3. Lines 230/237. Along the manuscript, there is a confusing discussion regarding the material degradation. For example, the authors mention that “After 15 days of implantation, the biomaterial still maintained its structural integrity with no breakdown”. However, Figure 2 shows a significant decrease in the mean biomaterial area. It is contradictory that the scaffold decreases the size without breakdown. A similar contradiction appears in “This reduction in thickness was steady with no disintegration” (line 399). It is necessary to clarify this point.
  4. Line 323. Have the authors found any correlation between the nature of the biomaterial and the possibility of absorbing the PRF? Is it related to the pore size or the physicochemical nature? It would be interesting for the readership to expand this discussion.
  5. Line 331. The authors state that “The matrix elicited only a mild physiological reaction on day 3 post implantation”. However, in the same paragraph, they say that “Only on day 30 was there significant rise in the density of the inflammatory cells when compared to days 3 and 15.” This is somehow contradictory. What do they refer to as “physiological reaction”?
  6. Line 368. Examples of each of the biomaterials included in Class I, Class II and Class III should be included.
  7. All figures. The size bars within the figures are difficult to read, please improve them.
  8. Misspelling: “hexamethylenediisocyanate” (line 57); multinucleated (lines 291 and 292).

Reviewer 3 Report

Manuscript ID: materials-811110

Review

General comment:

Dear Authors you presented a very interesting study including in vivo results regarding cellular response, vascularization and biodegradation of non cross-linked  collagen matrix. Recently the need for biomaterials is significantly rising and to be able to choose the right material to patients’ medical problem we need more information and understanding about biomaterials in vivo performance. Therefore, your manuscript presents a valuable work, however it requires some improvements.  

Detailed comments:

Line 22: I suggest using term “implantation site” instead “implantation bed” throughout the manuscript.

Lines 32-33: Please rewrite this sentence.

Lines 38:  “…do not meet all of these expectations..” please explain which particular expectations you meant.

Lines 45-46: Please rewrite this sentence.

Line 62: Should be “… type I collagen” instead of “type 1”.

Lines 81-84: You are describing already completed experiments use past tense instead of future tense.

Line 95: Please give the name of the supplier of Mucomaix®. Are there any reference supporting the conclusions made by manufacturers about their product you decribed?

Line 116: Please specify which histological analysis was done for ex vivo tested samples what was the protocol of staining?

Lines 125-126:  Please rewrite this sentence.

Line 138: “.. cut into three identical segments” you mean three segments from each sample?

Line 147: Please explain the abbreviation SMA.

Lines 166-169: This part you are repeating what was already written in lines160-162.

Lines 181-182:  Please rewrite this sentence.

Line 193: Fig. 1 A and B refers to untreated matrix only stained for imaging? Please describe this preparation in materials and methods.

Line 201: “i-PRF” means something special or it is only spelling mistake?

Lines 202-203: White areas in the matrix structure are the pores? How is it possible that some of the cells you marked in the Fig. 1D seems to be in the middle of pores, not attached to anything?

Lines 205-206: Is it really necessary to make a special paragraph for this one sentence?

Line 207:  Please specify “cellular reaction” to what.

Line 212-213: Can you show any image of control group for comparison?

Line 215: Try to unify scale bars at all images, image B’ is missing sale bar. Is there any particular reason why image A is higher magnification showing smaller area than images B and C? Graph should be marked as D (it also applies to other Figures containing graphs), at the graph you wrote “Mean biomaterial area” did you meant thickness? The description under the Fig.2 is very long and some of the information are repeated in the text consider to shorten it. Explain what is marked by blue dotted line. It seems quite surprising that you have statistical significance of matrix thickness between days at P<0.0001 with such high standard deviations.

Line 246: Scale bars are not readable. Here at graphs you wrote “3 days, 15 days, 30 days”, Fig 2. It was “3d, 15d, 30d” please unify it.

Lines 267-268:  Please rewrite this sentence.

Line 277: Can you show any image of the control sample? It seems you sacrificed 12 animals and have almost non results from control group.

Line 291: Should be “Multinucleated” instead of “Multinuleated”

Lines 299-306: You should write “percentage of vascularization”.

Lines 398-405: In the materials and methods as well as in the results you describe reduction of the thickness as one of the first results, so in the discussion it should be also written earlier.

Reviewer 4 Report

Dear Authors,

I have read the manuscript. Some questions raised. Enlisted please find my comments.

Overall. General English grammar revision (minor spelling errors).

Abstract. Please point out statistical analysis tests also in Abstract section.

Introduction. Authors stated “However, most of the available matrices do not meet all of these expectations, therefore necessitating a continued search for “the ideal” alternative”. Please rephrase as “However, most of the available matrices do not meet all of these expectations, therefore necessitating a continued search for an ideal alternative”.

Introduction. Clinical relevance could be stressed more. It could be pointed out that “Intra oral soft tissue augmentations are important procedures used in clinical periodontology. After healing, these procedures allow also the insertion of implants (Pérez-Albacete Martínez MÁ, Pérez-Albacete Martínez C, Maté Sánchez De Val JE, Ramos Oltra ML, Fernández Domínguez M, Calvo Guirado JL. Evaluation of a New Dental Implant Cervical Design in Comparison with a Conventional Design in an Experimental American Foxhound Model. Materials (Basel). 2018 Mar 21;11(4):462) or miniscrews (Sfondrini MF, Gandini P, Alcozer R, Vallittu PK, Scribante A. Failure load and stress analysis of orthodontic miniscrews with different transmucosal collar diameter. J Mech Behav Biomed Mater. 2018 Nov;87:132-137) with different design and shape”.

Materials and Methods. Authors stated “Mucomaix® is a resorbable matrix composed of highly purified porcine collagen and elastin fibers”. Please add manufacturer, city and state.

Materials and Methods. After introducing Mucomaix in the first sentence, please use the term “non cross-linked collagen-based matrix” (not commercial name) all along the Materials and Methods and Discussion sections. Also in the image captions commercial name should be avoided. If necessary use an abbreviation (Example: NCCM).

Materials and Methods. Authors stated “It is effectively purified and is therefore highly biocompatible and considered safe with regard to potential viral transmission”. Please add a reference for this statement.

Materials and Methods. Authors stated “has an open, interconnecting porous structure, which according to the manufacturers provides a matrix for the migration of proliferating cells and vascular structures”. A scheme or a drawing would be interesting for the readers.

Materials and Methods. Authors stated “They also claimed that as the healing process advances, Mucomaix® is degraded whilst a new soft tissue matrix is regenerated inside the matrix structure. Mucomaix® is hydrophilic and retains its structural integrity also in the wet state”. Please add some references for this statement. If no references are given it could be removed.

Materials and Methods. Authors stated “A very good method to assess this initial biomaterial-cell interaction is by the use of liquid-PRF”. Please rephrase as “A reliable method to assess this initial biomaterial-cell interaction is by the use of liquid-PRF”.

Materials and Methods. Authors stated “Three healthy volunteers aged between 18 and 60 years who gave written informed consent, donated blood for this ex vivo study”. Please state if and how sample size calculation has been performed. Additionally, please point out if an ethical committee approved the3 experimentation.

Materials and Methods. Authors stated “The liquid-PRF, which is the yellowish upper layer, was then collected into a syringe (BD 112 MicrolanceTM 3, Germany)”. Please add City of the manufacturer.

Materials and Methods. Authors stated “The 24 animals were first divided into 2 groups of 12 animals each”. Please state if and how sample size calculation has been performed.

Materials and Methods. Authors stated “These samples were placed into embedding cassettes (Histosette, VWR, Deutschland)”. Please add City of the manufacturer.

Materials and Methods. Authors stated “The tissue samples were then embedded into paraffin blocks and after sufficient cooling, the paraffin blocks were cut with a rotatory microtome (Rotationsmikrotom RM2255, Leica, Germany) to produce serial sections of 3-μm thickness”. Please add City of the manufacturer.

Materials and Methods. Authors stated “This  was done using the Lab VisionTM Autostainer 360-2D (ThermoFisher Scientific, Germany)”. Please add City of the manufacturer.

Materials and Methods. Authors stated “Proteinase K was blocked using 4% H2O2 in methanol and endogenous avidin- and biotin-binding proteins were blocked by the avidin and biotin blocking solutions (Avidin/Biotin Blocking Kit, Vector Laboratories, US)”. Please add City of the manufacturer.

Materials and Methods. Authors stated “The first antibody used was anti-CD-68 (MCA341GA; 1:400; 30 min), Anti-Human α-SMA antibody clone 1A4, whereas the second antibody was goat anti-rabbit IgG-B (sc-2040, 1:200, Santa Cruz Biotechnology, 153 USA)”. Please add City of the manufacturer.

Materials and Methods. Authors stated “Subsequently, the avidin- biotin-peroxidase complex (ABC) (30 min) and the Histostain-Plus 154 IHC Kit including AEC (20 min) were applied (ThermoFisher Scientific, Germany)”. Please add City of the manufacturer.

Materials and Methods. Authors stated “A Nikon ECLIPSE 80i microscope (Nikon, Tokyo, Japan) equipped with a motorized stage 160 (ProScan III, Prior, Rockland, MA, USA) and NIS Elements software (Nikon, Tokyo, Japan) was used 161 for qualitative histological analysis”. Please add magnification.

Materials and Methods. Authors stated “Statistical significance was calculated using one-way and two-way analysis of variance (ANOVA)”. ANOVA is used for Gaussian distributions. Please state if and how normality of data was tested.

Results. Authors stated “After 15 days of implantation, the biomaterial still maintained its structural integrity with no breakdown”. Please add significance (P value).

Results. Authors stated “Day 30 post-implantation showed that the matrix structure was still largely intact with no disintegrations, but there was however a significant reduction in the size of the biomaterial”. Please add significance (P value).

References. Some references are quite old (1991; 2001). If possible, please switch with some modern research. Some recent studies about the topic  have been suggested in the section above.

Round 2

Reviewer 4 Report

Good job